# The Role H-Bonding and Supramolecular Structures in Homogeneous and Enzymatic Catalysis

**DOI:** 10.3390/ijms242316874

**Published:** 2023-11-28

**Authors:** Ludmila I. Matienko, Elena M. Mil, Anastasia A. Albantova, Alexander N. Goloshchapov

**Affiliations:** N.M. Emanuel Institution of Biochemical Physics Russian Academy of Science, 4 Kosygin Str., 119334 Moscow, Russia; elenamil2004@mail.ru (E.M.M.); albantovaaa@mail.ru (A.A.A.); golan@sky.chph.ras.ru (A.N.G.)

**Keywords:** H-bonding, supramolecular structures, catalysis, AFM method, models of Oxygenases

## Abstract

The article analyzes the role of hydrogen bonds and supramolecular structures in enzyme catalysis and model systems. Hydrogen bonds play a crucial role in many enzymatic reactions. However, scientists have only recently attempted to harness the power of hydrogen bonds in homogeneous catalytic systems. One of the newest directions is associated with attempts to control the properties of catalysts by influencing the “second coordination sphere” of metal complexes. The role H-bonding, and the building of stable supramolecular nanostructures due to intermolecular H-bonds, based on catalytic active heteroligand iron (Fe) or nickel (Ni) complexes formed during hydrocarbon oxidations were assessed via the AFM (Atomic-force microscopy) method, which was proposed and applied by authors of this manuscript. Th is article also discusses the roles of hydrogen bonds and supramolecular structures in oxidation reactions catalyzed by heteroligand Ni and Fe complexes, which are not only effective homogeneous catalysts but also structural and functional models of Oxygenases.

## 1. Introduction

Hydrogen bond catalysis is a type of catalysis that relies on the use of hydrogen bonds to speed up and control organic reactions. In biological systems, hydrogen bonds often play a crucial role in orienting substrate molecules and lowering reaction barriers [1]. However, scientists have only recently attempted to exploit hydrogen bonding in homogeneous catalytic systems, and the field is relatively underdeveloped compared to research on, for example, Lewis acid catalysis [2]. Hydrogen bonds can stabilize anionic intermediates and transition states. Some catalysts can bind small anions, favoring the formation of reactive electrophilic cations.

Typically, the focus of scientific research is on the electronic and steric properties of axial modifying ligands, which affect the activity and selectivity of homogeneous metal complex catalysts [3]. The role of the second coordination sphere and hydrogen bonds in the mechanisms of catalytic reactions is less studied [4,5]. It was also noted in [6] that, recently, scientists have focused their attention on the “second coordination sphere”, i.e., on interactions beyond metal–ligand coordination that are important. These outer-sphere reactions involve non-covalent interactions between the ligands themselves, interactions with the ligand(s) and substrate(s), and/or interactions with the environment. This approach is likely inspired by natural enzymatic systems, since enzymatic conversions often depend on control by residues not directly involved in the chemistry of the catalytic cycle.

The weak outer-sphere interactions involving hydrogen bonds often play a structurally organizing role in enzymatic and homogeneous catalytic processes [7].

The chemical industry is one of the main sources of environmental pollution. Therefore, fundamentally new chemical processes with a reduced level of energy intensity and minimal formation of undesirable by-products are needed. The development of industrial processes for the oxidation of hydrocarbons is associated with the ability to control these processes. An effective method to control the rate and mechanism of hydrocarbon oxidation is the use of a catalyst. In recent years, research in the field of hydrocarbon oxidation with molecular oxygen has been developing in two directions: free radical chain oxidation catalyzed by transition metal complexes and catalysis by metal complexes that imitate the action of enzymes [3].

A common disadvantage of the most biomimetic systems is the low yields of oxidation products of 1–2% per consumed hydrocarbon due to the rapid deactivation of the catalyst, which is the main obstacle to the use of these systems on an industrial scale. In addition, scientists have used the mechanisms of action of known dioxygenases, which catalyze the oxidation of a limited number of relatively active substrates, mainly the regioselective oxidation of easily oxidizable hydrocarbons—alcohols, aldehydes. Unfortunately, dioxygenases capable of carrying out chemical reactions of alkane dioxygenation are unknown.

Conclusions about the mechanisms of action of dioxygenases and their models can be used to interpret the mechanisms of the oxidation of hydrocarbons by molecular oxygen catalyzed by metal complexes.

Matienko L.I. and others. were the first to propose a method for modifying metal complexes with electron-donating activating ligands in order to increase the selectivity and conversion of the homogeneous catalytic oxidation of alkylarenes (ethylbenzene and isopropylbenzene) with molecular oxygen into the corresponding hydroperoxides [3]. (Hydroperoxides are intermediates in the large-scale production of important monomers. α-Phenylethyl hydroperoxide is an intermediate in the synthesis of propylene oxide and styrene, and cumyl hydroperoxide is an intermediate in the synthesis of phenol and acetone [3]). The mechanism of the formation of active forms of catalysts (due to reactions in the outer coordination sphere of metal complexes initiated by electron-donating ligands) has been established, and new highly efficient catalysts (nickel complexes) for the selective oxidation of ethylbenzene into α-phenylethylhydroperoxide (PEH) have been designed [3]. L.I. Matienko suggested that the excellent efficiency and stability of active catalytic complexes in the oxidation process might be associated with the formation of supramolecular structures due to intermolecular H-bonds or, possibly, other non-covalent interactions [3]. This hypothesis was confirmed by using the AFM method (atomic force microscopy), and a new approach to studying the role of H-bonds and supramolecular structures in the mechanisms of homogeneous and enzymatic catalysis (the AFM method) was proposed [8,9].

## 2. Results and Discussion

### 2.1. The Role of the Secondary Interactions in Homogeneous Mechanisms and Enzymatic Catalysis

#### 2.1.1. H-Bonding and the O_2_ Activation

H-bonds are defined by the IUPAC as follows: “An H-bond is a form of association between an electronegative atom and a hydrogen atom attached to a second, relatively electronegative atom.” H-bonding is best viewed as an electrostatic interaction enhanced by the small size of hydrogen, which ensures the proximity of the interacting dipoles or charges [10].

Pauling’s definition of H-bonding is as follows: “Under certain conditions, atom of hydrogen is attracted by rather strong forces to two atoms, instead of only one, so that it may be considered to be acting as a bond between them. This is called hydrogen bond.” Pauling also states that the hydrogen bond “is formed only between the most electronegative atoms.” The terms “typical hydrogen bond” and “conventional hydrogen bond” are related in the review [10], which refers to Pauling’s definition of H-bonds.

H-bonds have a wide range of energies: very strong ~ 62.7–167.2 kJ/mol; weak—16.7 kJ/mol. These are weaker bonds compared to typical chemical bonds [10]. For example, the breaking of ionic bonds (Na^+^Cl^−^) requires 788 kJ/mol of energy, while the breaking of covalent bonds (C-H bonds) requires 411 kJ/mol. The weakest interaction is characteristic of CH_4_•••FCH_3_ (energy ~ 0.8 kJ/mol) [10]. Much of the calculation suggests that strong hydrogen bonds are more covalent in nature. Electrostatics may be more important for weak hydrogen bonds. The contribution of electrostatics to hydrogen bonding has been widely discussed [11]. Hydrogen bonds play an important role in non-covalent aromatic interactions, where π electrons play the role of proton acceptor, which is a very common phenomenon in chemistry and biology. H-bonds play an important role in the structures of proteins and DNA, as well as in the binding and catalysis of drug receptors [12]. Topping the list as the earliest and still the most studied systems are proton-linked bicarboxylates, proposed to form low-barrier hydrogen bonds (LBHB) near the active sites of enzymes [13]. Spin–spin coupling between all three nuclei of a hydrogen bridge was observed for the first time. The observation of these couplings in the hydrogen fluoride/collidine complex reveals a covalent character of the hydrogen bridge. Its geometry is strongly affected by temperature-dependent solvent electric field effects [14].

In 1912, Alfred Werner put forward the idea that the microenvironment of variable-valence metal complexes influences the structure and function of the complexes [15]. In [15,16], the authors study the influence of intramolecular H-bonds in the second coordination sphere of metal complexes on their ability to bind O_2_ and activate O_2_.

Secondary interactions (hydrogen bond, proton transfer) play an important role in the activation of O_2_ when it binds to the active center of metalloenzymes [17]. Respiration becomes impossible if the fragments responsible for the formation of H-bonds with Fe-O_2_ in the active center are removed from the active center of hemoglobin [18]. In addition, the affinity of the active sites of hemoglobin for O_2_ is in a certain relationship with the network of H-bonds surrounding the Fe ion. The dysfunction of cytochrome P450, which is observed when the H-bonds formed by the Fe-O_2_ fragment are broken, indicates the important role of H-bonds forming the second coordination sphere around the metal ions of many proteins [19].

#### 2.1.2. Reactions in the Second Coordination Sphere of Metal Complexes including O_2_ Addition, Proton or Hydrogen Transfer, H-Bonding

A systematic study of the interactions leading to the formation of outer-sphere complexes (OSCs) was carried out by Kitaygorodsky A.N., Nekipelov V.M., and Zamaraev K.I. in 1976 during the study of tris-β-diketonates of various transition metals. Molecules that are proton donors, even if they are very weak (acids, alcohols, alkaloids, etc.), can form OSCs through hydrogen bonds with intra-sphere atoms and ligands. At the same time, molecules such as acetone, acetonitrile, benzene, and toluene, which do not have proton donor properties, apparently form OSCs through the interaction of their π-systems with the π-systems of inner-sphere ligands [20].

The formation of OSCs can significantly change the physicochemical properties of metal complexes, such as their solubility. In addition, OSC formation can have a significant effect on the rate of transformations of intra-sphere ligands and the transfer of the H atom between different groups of intra-sphere ligands [20].

In some cases, the orientation of reagents in the OSC of complexes can promote transformations catalyzed by metal complexes (for example, the formation of urethanes in the coordination sphere of the Fe(acac)_2_OR complex, where OR is an alkoxide anion, as well as the formation of semi-acetals from alcohols and b-diketonate ligands coordinated by Fe(III) [20]. The “Synchronous transfer of a proton or a hydrogen atom along a chain of chemical and hydrogen bonds in cyclic intermediates may be important for a wide class of organic reactions occurring in the coordination sphere of metal complexes” [7]. The weak outer-sphere interactions often play a structurally organizing role in these processes. Like enzymatic catalysis, hydrogen bonds play an important role in the transfer of a proton or a hydrogen atom along bond chains [7,20].

The *β*-Diketonates of transition metals can participate in various outer-sphere substitution reactions. The various electrophiles (E) can replace methine protons of chelate rings in *β*-diketonate ligands (formally, these reactions are similar to Michael addition reactions) [21,22]. This is a metal-controlled C-C bond formation process [21]. The most effective catalyst in such reactions is Ni(acac)_2_. The formation of the resonance-stabilized zwitterion [(M^II^L^1^_n_)^+^E^−^] is the rate-determining step of these reactions. Proton transfer in a zwitterion precedes the formation of reaction products [3]. The isolation of Ni(L^1^•E)_2_ complexes from the reaction medium confirms the role of the metal in ligand activation in the E addition reaction [22].

In the works of Matienko L.I. et al., the efficient catalytic systems {ML^1^_n_ + L^2^} (M=Ni, Fe, L^2^—activating the axial electron-donating ligand) for the oxidation of ethylbenzene to α-phenyl ethyl hydroperoxide have been modeled. These catalytic systems are constructed based on mechanisms (established for Ni complexes and hypothetical for Fe complexes) of the formation of active and excellent effective catalytic species in the oxidation process. These active catalytic complexes are formed in the process of the oxidation of ethylbenzene as a result of the outer-sphere regioselective addition of O_2_ to the L^1^ ligand of the M(L^1^)_2_•L^2^ complexes. Based on the experimental evidence obtained, it was supposed that intermolecular H-bonds and the self-assembling supramolecular structures may play an important role in this mechanism [3].

The two types of active complexes formed during ethylbenzene oxidation, catalyzed by systems {ML^1^_n_ + L^2^} (M=Ni, L^2^ = crown ethers or quaternary ammonium salts, as well as various monodantate ligands, L^1^ = acac^−^, or enamac^−^). These are the primary (M^II^L^1^_2_•L^2^) complexes formed in the beginning stages of ethylbenzene oxidation (I stage of catalytic process) and the heteroligand M^II^*_x_*L^1^*_y_*(L^1^_ox_)_z_(L^2^)*_n_*(H_2_O)*_m_* complexes formed in the oxidation process (L^1^_ox_ = intermediate product of L^1^ oxidation by O_2_) (II stage). The conversion of complexes to the final product M(OAc)_2_ leads to a decrease in the selectivity of ethylbenzene oxidation in PEH (III stage).

The general kinetic patterns of ethylbenzene oxidation suggest a commonality in the mechanisms of formation of actual catalysts M*_x_*L^1^*_y_*(L^1^_ox_)_z_(L^2^)*_n_*(H_2_O)*_m_* for nickel and iron complexes. The coordination of the axial electron-donating ligand L^2^ leads to the formation of active ML^1^_2_•L^2^ complexes and promotes subsequent outer-sphere transformation reactions of β-diketonate ligands upon interaction with molecular oxygen. The coordination of the extra electron-donating ligand L^2^ favors the stabilization of the zwitter ion L^2^[L^1^M(L^1^)^+^O_2_^−^], increasing the likelihood of the regioselective addition of O_2_ to the methine C−H bond of the acetylacetonate ligand, which is activated by its coordination with the metal ion. The subsequent stages, namely the outer-sphere reaction of O_2_ insertion into the chelate ring, depend on the nature of the metal and the modifying ligand L^2^ [3].

For nickel complexes Ni^II^_x_L^1^_y_(L^1^_ox_)_z_(L^2^)_n_, the oxygenation reaction of the acac ligand occurs according to a mechanism most likely similar to the action of Ni-containing Acireductone Dioxygenase (NiARD) [23] or Cu- and Fe-quercetin 2,3-Dioxygenases [24,25]. The inclusion of O_2_ into the acac chelate ring, proton transfer, and rearrangement of bonds in the transition complex and subsequent cleavage of the cyclic configuration results in the formation of the chelating ligand OAc-, acetaldehyde, and CO (via the Criegee rearrangement).

For iron catalysis. A favorable combination of electronic and spherical factors during the intra- and outer-sphere (hydrogen bonding) coordination of the L^2^ ligand to Fe(acac)_2_ may favor a different route, according to the mechanism realized during the action of Fe^II^ARD [22] or Fe^II^-acetyl acetone Dioxygenase (Dke1) [26]. This hypothetic mechanism includes O_2_ activation [26]. In the next stage, oxygen is added to the C-C bond (rather than incorporated into the C=C bond as in catalysis by nickel(II) complexes) to afford an intermediate complex with a ligand containing 1,2—a dioxetane fragment. The reaction products are as follows: (OAc)^−^ chelate ligand and methylglyoxal (see Figure 1, wherein the complex [Fe(acac)_2_•18C6] acts as an example):

The important role of H-bonds in the formation of active forms of catalysts was established via the addition of small amounts of water (~10^−3^ mol litre^−1^) to the {ML^1^_n_ + L^2^} (M=Fe) systems. It was discovered that the addition of small amounts of water leads to an increase in the initial rates of RH oxidation and an increase in the parameter S_PEH_ in the course of the reaction [3,27,28]. The observed decrease in the parameter (S_PEH_)_max_ in the case of catalysis by complexes Fe(acac)_2_•18C6 might be explained by the decrease in the stationary concentration intermediates Fe_x_^II^(acac)*_y_*(OAc)*_z_*(18C6)_n_(H_2_O)_m_ due to the acceleration of the conversion of these reactive intermediates into the final product Fe(OAc)_2_. It is known that the addition of low concentrations of water (~10^−3^ mol litre^−1^) to hydrocarbons does not disturb the homogeneity of the medium [3].

After our work in article [29], it was found that the probability of transforming the β-diketone ligand of the iron complex via analogy with the action of FeARD or Fe-Dke-1 increases in an aquatic environment. This is consistent with the data obtained in our previous works [27,28].

The interaction of enzyme molecules with water molecules is of critical importance for enzymatic activity [30]. The water molecules present in the active site of a protein can not only play a structuring role (as a nucleophile and proton donor) but also may be considered reagents in biochemical processes. Thus, the first step in breaking the O-O bond in the H_2_O_2_ molecule (in Fe-horseradish peroxidase) is the transfer of a proton to the histidine residue (His42). This step is simplified if a water molecule is present in the active site of the peroxidase [31]. According to ab initio calculations, the energy barrier of this reaction is significantly higher (83.6 kJ/mol) in the absence of a water molecule compared to its experimentally found value (6.3 ± 2.9 kJ/mol).

The role of hydrogen bonds in the mechanism of action of Heme oxygenase (HO), a metal-free enzyme whose ligand in HO is water, has been studied [32].

Changes in water concentration have different effects on the intra- and extra-diol oxygenation of 3,5-di(tert-butyl)catechol with dioxygen in a water tetrahydrofuran in the presence of FeCl_2_ or FeCl_3_. It was assumed that various intermediates are formed in these reactions. Extradiol oxygenation occurs only with Fe^2+^ salts, in contrast to the intradiol mechanism with Fe^3+^ salts (catechol 2,3-dioxygenase model) [30,33].

### 2.2. Nanostructure Science and Supramolecular Chemistry

Nature is exploiting non-covalent interactions for the construction of various cell components. Structures such as microtubules, ribosomes, mitochondria, and chromosomes use hydrogen bonding, along with covalent peptide bonds, to form specific structures.

Nanostructure science and supramolecular chemistry are rapidly evolving fields that are concerned with the manipulation of materials that have important structural features of nanometer size (1 nm to 1 μm) [34,35].

Nobel laureate Lehn Jean-Marie first coined the term “supramolecular chemistry” in 1978: “Supramolecular chemistry”—“chemistry describing complex formations that result from the association of two (or more) chemical particles bound together by intermolecular forces.” “Supramolecular chemistry is the “chemistry beyond the molecule,” the study of the structure and function of associations of two or more chemical particles held together by intermolecular forces.” (J.-M. Len, 1987: Introduction of the concepts of “self-organization” and “self-assembly”) [36].

The following years were marked by the explosive development of this young interdisciplinary science. In 2016, Sirs James Fraser Stoddart, Jean-Pierre Sauvage, and Bernard Feringa received the Nobel Prize in Chemistry for their research in this area.

“Supramolecular chemistry has implemented principles of molecular information in chemistry. Through manipulation of intermolecular noncovalent interactions, it explores the storage of information at the molecular level and its retrieval, transfer, and processing at the supramolecular level via interactional algorithms operating through molecular recognition events based on well-defined interaction patterns (such as hydrogen bonding arrays, sequences donor and acceptor groups, and ion coordination sites)” [36]. “Supramolecular chemistry provides ways and means for progressively unraveling the complexification of matter through self-organization. Together with the corresponding fields in physics and biology, it leads toward a supramolecular science of complex, informed, self-organized evolutive matter” [36].

Based on the several attempts in the literature to define the term “self-assembly”, the authors of [37] outlined the following commonalties for metal-mediated supramolecular self-assembly. (a) Self-assembled units are held together through coordination interactions. (b) The assembly of subunits into large aggregates is selective: subunits form the most stable aggregates. (c) Aggregates differ in their properties from the properties of individual components. (d) Aggregates are discrete, not infinite. Self-organizing supramolecules are generally thermodynamically preferable to oligomeric or polymeric systems because they have both enthalpic and entropic effects.

In [37], self-assembly is considered as a subset of self-organization. It is agreed in [37] that self-organization results in no discrete systems with varying sizes, where a missing component does not alter the structure, and the properties are not affected below a critical defect density. These are often topologically open systems (for example, coordination polymers, aggregates, crystals, and monolayers).

Specific intermolecular interactions [37] are directional motifs designed to recognize complementary components to yield predictable intermolecular interactions, such as H-bonds, metal ion coordination, and dipolar interactions. These generally lead to predictable self-assembled architectures in solution. Nonspecific intermolecular interactions are generally nondirectional, such as dispersion forces and ionic interactions. These generally lead to structures that are difficult to predict.

Self-complementarity [37] means specific intermolecular interactions between copies of the same groups, for example, H-bonds between a carboxylic acid dimer. Heteroclementary implies specific intermolecular interactions between two different groups.

H-bonds are commonly used to make supramolecular assemblies because they are directional and have a wide range of interactions that can be exploited by adjusting the number of H-bonds, their orientation, and their position in the overall structure. Due to cooperative dipolar interactions, H-bonds in the center of protein helices can reach 20 kcal/mol [36].

Porphyrin bonding via H-bonds represents the type of bonding observed in nature. One of the artificial self-organizing supramolecular porphyrin systems is the formation of a dimer based on carbon functionality [37,38] (Figure 1 [38]):

Mn(salen) complexes are used as catalysts for enanthoselective epoxidation. The modification of Mn(salen) by forming a supramolecular complex with zinc porphyrinate (due to the H-bond) increases the stability of the catalyst and promotes an increase in the substrate selectivity of the reaction [38] (Figure 2):

#### 2.2.1. Models Ni(Fe)ARD Dioxygenases

Reactions of the cleavage of C–C bonds in β-diketones by enzymes are of interest in connection with the study of various aspects of cells, biocatalysis, and the physiology of mammals. The mechanisms by which C–C bonds are broken are extremely diverse, ranging from hydrolytic processes involving metals to reactions catalyzed by dioxygenases [3].

The methionine salvage pathway (MSP) plays an important role in the regulation of a number of metabolites in prokaryotes and eukaryotes. The MSP is a universal pathway for the conversion of sulfur-containing metabolites into methionine [39]. The two Ni(Fe)ARD enzymes are members of the cupin structural superfamily, which includes Fe-Acetylacetone dioxygenase (Dke1) and cysteine dioxygenase. These enzymes use a triad of histidine ligands (His), as well as one or two oxygen atoms from water and carboxylate oxygen (Glu), to bind to the Fe(Ni) center [40].

Two Ni(Fe)ARD enzymes have the same amino acid sequence and differ only in the metal ions. NiARD and FeARD act on the same substrates—Acireductone, 1,2-dihydroxy-3-keto-5-methylthiopentene anion, and O_2_—but produce different products. FeARD catalyzes the oxidation of Acireductone to formate and 2-keto-4-methylthiobutyrate, which is a precursor to methionine [22]. The reaction pathway catalyzed by NiARD does not produce methionine, but the reaction does produce CO, which is a neurotransmitter. CO has been identified as an antiapoptotic molecule in mammals (mouse and human ARD isozymes) (Figure 2 [39]). A study of the influence of metal ions Fe or Ni, as well as Co and Mn, on the catalytic activity of the mammalian enzyme showed that Fe-ARD, which catalyzes the first pathway, exhibits 10 times higher activity compared to Ni-, Co-, and Mn-containing enzymes that catalyze side reaction pathways. Ni^2+^MmARD exhibits the greatest thermal stability, followed by Co^2+^ and Fe^2+^, and Mn^2+^ARD has the least stability [41]. In addition to its enzymatic function, studies have shown a role for the ARD enzyme in carcinogenesis and tumor metastasis. Thus, it was recently discovered that the human enzyme regulates the activity of matrix metalloproteinase I (MMP-I), which is involved in tumor metastasis, by binding to the cytoplasmic transmembrane tail peptide MMP-I [39].

ARD is also implicated in hepatitis C infection, Down’s syndrome (DS)-associated congenital heart defects, and fecundity in Drosophila. In plants, ARD expression is associated with development and fruit ripening and is therefore of great interest to plant biologists. ARD is likely a multi-functional (“moonlighting”) enzyme involved in both regulatory and enzymatic functions [41].

We hypothesized that one of the possible reasons for the different activity of Ni(Fe)ARD towards common substrates (Acireductone and O_2_) may be the self-organization of catalysts into different supramolecular structures due to intermolecular H-bonds.

The action of FeARD apparently includes the stage of the activation of oxygen (Fe^II^ + O_2_ → Fe^III^ − O_2_^−·•^) (by analogy with the action of Dke1 [26,40]). The specific structural organization of iron complexes may favor the regioselective addition of activated oxygen to the Acireductone ligand and reactions leading to the formation of methionine. Self-organization in stable macrostructures may be one of the reasons for the decrease in NiARD activity in the mechanisms of Ni(Fe)ARD operation.

Using the AFM method, we studied the possibility the self-organization of iron (nickel) heteroligand complexes in supramolecular structures due to intermolecular H-bonds [9].

Figure 3 presents a 2D AFM image of the supramolecular structures based on the iron complex Fe^III^*_x_*(acac)*_y_*18C6*_m_*(H_2_O)*_n_*, formed via putting a uterine solution on a hydrophobic surface of modified silicone. As you can see, the formed structures are organized in a certain way, forming structures resembling the shape of a tubular microfiber cavity (Figure 3c). The heights of the particles are about 3–4 nm.

It is known [3] that heteroligand complexes in reactions with electrophiles are characterized by higher activity compared to homoligand complexes. In [3], we assumed that the stability of model heteroligand nickel complexes Ni*_x_*(acac)*_y_*(OAc)*_z_*(L^2^)_n_(H_2_O)_m_ (Ni_2_(OAc)_3_(acac)NMP• 2H_2_O, L^2^ = NMP) in oxidation reactions of alkylarenes may be associated with the formation of stable supramolecular structures due to intermolecular H-bonds. This assumption was confirmed by AFM microscopy data [42].

As you can see in Figure 4, using AFM, we observed the self-organization of model binuclear supramolecular complexes Ni_2_(AcO)_3_(acac)·MP·2H_2_O into supramolecular nanostructures due to intermolecular H-bonds (H_2_O—MP, H_2_O—(OAc^−^)(or (acac^−^)). All structures are on various heights ranging from a minimum of 3–4 nm to a maximum of ~20–25 nm (in a form resembling three almost merged spheres). The structure of complex Ni_2_(AcO)_3_(acac)·MP·2H_2_O (Figure 4a) was defined by mass spectrometry, electron and IR spectroscopy, and element analysis [42].

#### 2.2.2. Possible Effect of the Tyr Fragment in the Second Coordination Sphere of Ni(Fe)ARD

We suggested that it is necessary to take into account the role of the second coordination sphere, in particular the Tyr fragment, in the functional mechanisms of Ni(Fe)ARD enzymes [42] (Figure 3).

In [42], we analyzed the participation of tyrosine in various enzymatic reactions [43,44,45].

So, for example, the role of tyrosine residues in the mechanism of action of heme oxygenase (HO) has been studied. H O is responsible for the cleavage of histidine-ligated trivalent protoporphyrin IX (Por) to biliverdin, CO, and free iron ions. Tyrosyl radical formation after the oxidation of Fe(III)(Por) to Fe(IV)=O(Por(·+)) in human heme oxygenase isoform-1 (hHO-1) and the structurally homologous protein from Corynebacterium diphtheriae (cdHO) are described in [43].

Tyr-fragment may take part in step of O_2_-activation due to H-Bonding. As assumed this interaction can lead to decrease the oxygenation rate of the substrate in the case of catalysis with Homoprotocatechuate 2.3-Dioxygenase [44]. Tyr-fragment takes part in methyl group transfer from S-adenosylmethionine (AdoMet) to dopamine [45]. Methyl CH•••O hydrogen bonding (with participation of Tyr-fragment) was assumed to represent a feature of AdoMet-dependent Methyltransferases as the universal mechanism for methyl transfer [46].

We assumed that the Tyr fragment, which is located in the outer sphere of NiARD (Figure 3), could reduce the activity of the Nickel enzyme.

The phenomenon of increasing the selectivity (**S**) and conversion (**C**) of the ethylbenzene oxidation to PEH upon the addition of PhOH, together with the electron-donating ligands L^2^ (NMP, HMPA or MSt (M = Li, Na)), to the metal complex Ni(acac)_2_ was discovered in the works of L.I. Matienko and L.A. Mosolova [3]. In the case of {Ni(acac)_2_ + NaSt + PhOH}, the values of conversion **C** (>35% at **S**_PEHmax_ ≅ 90%), [PEH]_max_ = 1.6−1.8 mol/L, far exceeded those obtained with the other triple catalytic systems and the majority of active binary systems. The Russian Federation patent (2004) protects these results reported by L.I. Matienko and L.A. Mosolova. A distinctive feature of these ternary systems is that the resulting Ni(acac)2•L2•PhOH complexes are not transformed by RH oxidation and have long-term activity. Unlike binary systems, the acac- ligand in the ternary nickel complex is not transformed during the oxidation of ethylbenzene. (The formation of ternary complexes {Ni(acac)_2_∙L^2^∙PhOH} at the earliest stages of oxidation was established using kinetic and spectral methods [3,8]). Kinetic methods have been used to establish the role of intramolecular H-bonds in the mechanism of the formation of ternary complexes ({Ni(acac)_2_∙L^2^∙PhOH}, L^2^ = NMP) during the oxidation of ethylbenzene with molecular oxygen [3,8].) The dual function of phenol, coordinating with the nickel complex, was discovered to be dependent on the ligand environment of the metal (nickel). As previously established, {Ni(acac)_2_∙PhOH}- complexes are effective antioxidants in the oxidation of RH by molecular O_2_, and ternary complexes {Ni(acac)_2_∙L^2^∙PhOH} are effective catalysts for the selective oxidation of RH to PEH [47].

Apparently, the inclusion of phenol into the {Ni(acac)_2_∙L^2^} complex and the formation of triple complexes {Ni(acac)_2_∙L^2^∙PhOH} prevents the introduction of oxygen through the C-H-methine bond of the ligand acac^−^ and oxidation transformation of Ni complex. The formation of supramolecular structures due to H-bonds (phenol–carboxylate) and possibly other non-covalent interactions facilitated the stabilization of triple complexes with respect to the reaction with molecular oxygen.

The AFM microscopy data we obtained (Figure 5) evidence the formation of stabile supramolecular nanostructures based on triple complexes {Ni(acac)_2_∙L^2^∙PhOH} during the real catalytic oxidation of ethylbenzene. The self-organization of triple complexes on surfaces of modified silicon (AFM) are driven by the balance between intermolecular and molecule–surface interactions, which may be the consequence of hydrogen bonds and other non-covalent interactions [47,48,49,50,51]. So, for example, L-leucine derived ligand (H(2)L(L-leu)), KOH, and Ni(II) salt in 2:2:1 ratio self-assembled into a rather large (approximately 13 A) supramolecular assembly with the formula [K{Ni(HL(L-leu))(2)}(3)](+) (1). Structural characterization showed three [Ni(HL(L-leu))(2)] units encapsulated K(+) similar to organic crown ethers/cryptand. Electrospray Ionization (ESI)-mass spectra of the assemblies in MeOH showed the retention of assemblies in solution [48]. In [50] [Ni(2)(L)(2)(OAc)(2)] (1) and [Ni(3)(L)(2)(OAc)(4)(H(2)O)(2)].CH(2)Cl(2).2CH(3)OH (2) have also been isolated and structurally and magnetically characterized. The structural analysis reveals that the Ni(II) ions possess a distorted octahedral geometry formed by the chelating tridentate ligand (L), a chelating acetate ion, and a bridging phenoxo group with very similar bond angles and distances (HL = the tridentate Schiff base ligand).

We obtained similar data in the case of the use of L-Tyrosine (Tyr) as an extra ligand. Using the AFM method on the surfaces of modified silicon, we observed, for the first time, self-assembled supramolecular structures based on the triple systems {Ni(acac)_2_ + NMP(His) + Tyr} due to intermolecular (phenol–carboxylate) H-bonds and, possibly, due to other non-covalent interactions (Figure 6). UV spectroscopy data indicate the intra- and outer-spherical coordination of extra ligands Tyr and His with Ni(acac)_2_ [9].

In Figure 7, one can see the self-organization of triple iron complexes, including Tyr and His as activating ligands, Fe^III^*_x_*(acac)*_y_*(His)*_m_*(Tyr)*_n_*(H_2_O)_p_ (B) into tubulin-like structures due to H-bonds. Nickel complexes do not form such structures.

It is quite possible that in the case of the formation of (B) triple complexes, the Tyr fragment does not lead to a decrease in the activity of the enzyme. On the contrary, the outer-sphere coordination of the Tyr fragment may favor the formation of tubulin-like structures and O_2_ activation (due to H-bonds), as well as the subsequent regioselective addition of activated oxygen to the acac ligand and reactions leading to the formation of methionine.

Apparently, the presented AFM data can contribute to the understanding of the processes occurring as a result of the action of Ni(Fe)ARD and the role of the Tyr fragment in the synthesis of methionine and CO, as well as in the mechanism of normal homeostasis.

#### 2.2.3. Models of Family Fe-Mono Oxygenases P450: The Possible Role of Tyrosine and Histidine Fragments

The family of cytochrome P450-dependent monooxygenases is part of a class of hemoproteins with extremely diverse functions. Scientists have used advances in the fields of biology, biophysics, computational chemistry, and spectroscopy to gain information about the influence of various structural factors on the reactivity of the active sites of proteins. Studies have shown that the coordination spheres of the active centers of proteins play a decisive role in determining the properties of the metal cofactor. The importance of H-bonds for the coordination of molecular oxygen O_2_ and activation of O_2_ by P450 metalloenzymes has been studied [52,53,54,55]. One approach to understanding their biological function is to study synthetic constructs that mimic one or more aspects of native metalloproteins [53].

In nature, various types of binding of amino acids to Metalloporphyrins are observed. For example, it has been found that TYR (tyrosine) can form hydrogen bonds with HIS (histidine) [52,53,54,55,56].

For the first time, we used the AFM method to study the possibility of forming supramolecular structures based on metal complexes of porphyrin with amino acids, tyrosine, and histidine. These complexes are part of the active centers of enzymes, in particular cytochrome P450-dependent monooxygenases. The formation of such structures can be used to analyze non-covalent intermolecular interactions in the mechanisms of enzymatic catalysis [55].

We expected to observe the self-organization of supramolecular structures due to H-bonds based on complexes of iron porphyrins with tyrosine and histidine, which are models of P450 enzymes. This assumption would have been based on data in the literature on the regulatory role of histidine and tyrosine fragments in the functioning of the enzymes (heme proteins) of the P450 family [56]. As mentioned earlier, the binding of porphyrins via H-bonds is a common intermolecular interaction in nature [37,38].

Figure 8a,b show AFM images of stable nanostructures based on model systems {Hem + Tyr + His} in the form of triangular prisms. These structures are apparently formed as a result of the spontaneous self-organization of {Hem•Tyr•His} complexes due to intermolecular interactions involving hydrogen bonds and, possibly, other non-covalent interactions [42,57].

One can see in Figure 8c one of the possible schemes for the formation of triangular complexes based on the {Hem + Tyr + His} system (and then prisms) due to H-bonds (NH•••O or N•••HO) [55,56,58]. The combination of the individual planar {Hem·Tyr·His} complex (Figure 8c) with Sierpinski triangular motifs [59] into triangular structures (Figure 8d) with the participation of H-bonds and then π–π- intermolecular interplanar interactions can lead to the formation of triangular prisms (Figure 8a,b).

## 3. Materials and Methods

Ethylbenzene (RH=C_6_H_5_CH_2_CH_3_) was oxidized by O_2_ at 120 °C in a glass bubbling-type reactor in the presence of catalytic systems. The analysis of the oxidation products: *α*-Phenyl ethyl hydroperoxide (PEH=C_6_H_5_HCOOH(CH_3_)) was analyzed via iodometry. The byproducts methylphenylcarbinol (MPC=C_6_H_5_HCOH(CH_3_)), acetophenone (AP=C_6_H_5_CO(CH_3_)), phenol (PhOH=C_6_H_5_OH), and also RH during the oxidation reaction, were examined using GLC [8]. Experimental data processing was performed using the special computer programs Mathcad and Graph2Digit.

For the AFM study, we used the scanning probe microscope SOLVER P47 SMENA10 (Adm. Distr. Zelenograd of Moscow, 124490 Moscow, Russia) at a frequency of 150 kHz using an NSG30 cantilever with a radius of curvature of 10 nm. We used an NSG30_SS cantilever (Nanosensors^TM^ Advanced Tec^TM^ AFM probes, CH-2000 Neuchatel, Switzerland) with a radius of curvature of 2 nm, a resonance frequency of 300 kHz, and a force constant of 22–100 N/m. TipsNano in tapping mode was used for the AFM study of supramolecular structures. Sampling was carried out using a spin coating process, from H_2_O, CHCl_3_ solution (Ni(acac)_2_∙L^2^∙PhOH(Tyr)) complexes, or H_2_O-C_2_H_5_OH solution {Hem·Tyr·His} complexes). Measurements were conducted with air-dried samples. AFM SOLVER P47/SMENA was used with Silicon Cantilever NSG11S (NT MDT), which has a curvature radius of 10 nm, a 2 nm tip height, 10–15 µm size, and a cone angle ≤ 22°. It was used in taping mode at a resonant frequency of 150 KHz.

UV spectroscopy was used to prove the roles of the Tyr and His fragments in the formation of nickel complexes. Quartz cuvettes, which were 1 mm thick, were used to record the spectra in the UV regions. The spectra were recorded on a high-sensitivity spectrophotometer, “UV-VIS-SPECS”.

## 4. Conclusions

When studying the properties of homogeneous metal complex catalysts, scientists usually pay attention to the steric and electronic properties of the ligands, which can change the activity of the catalysts. In this regard, interactions in the external coordination sphere, the role of hydrogen bonds, and the role of other non-covalent interactions are much less studied. One of the latest strategies for controlling catalyst properties is targeting the “second coordination sphere,” that is, important interactions beyond metal–ligand coordination.

The mechanism of catalysis often involves the formation of supramolecular structures due to non-covalent interactions during the catalytic reaction. Supramolecular structures open up a very promising direction for the analysis of non-covalent interactions.

We have proposed an original approach: the use of atomic force microscopy (AFM) to assess the role of intermolecular hydrogen bonds and supramolecular structures in the mechanisms of homogeneous and enzymatic catalysis. The developed methodological AFM approach made it possible to confirm the conclusions based on kinetic studies about the possible self-organization of effective forms of catalysts (heteroligand Ni and Fe complexes) into supramolecular structures due to intermolecular H-bonds (and, possibly, due to other non-covalent interactions) in the processes of the selective oxidation of RH by O_2_. The formation of such supramolecular structures may be one of the reasons for the stability and high efficiency of catalysts. The self-organization of supramolecular structures based on heteroligand complexes of nickel and iron on the surface (AFM method) indicates a high probability of the formation of such structures under real conditions of catalytic oxidation.

The possible roles of supramolecular structures and hydrogen bonds in the mechanisms of enzymatic catalysis discussed in our works are due to the similarity of the mechanisms of formation of effective forms of Ni and Fe catalysts and the mechanisms of action of Ni(Fe)ARD Acireductone Dioxygenases. There is also a similarity between the structures of active catalysts and the structures of active sites of Ni(Fe)ARD enzymes. The role of the second coordination sphere and the role of the Tyr fragment in the mechanism of action of Ni(Fe)ARD have been discussed.

AFM data based on nickel and iron complexes modeling the active centers of Ni(Fe)-ARD, as well as enzymes of the P450 family, are of particular interest from the point of view of assessing the possible role of supramolecular structures and Tyr and His fragments in the mechanisms of enzymatic catalysis.

## 5. Future Directions

We suggest that the AFM method is one of the most useful methodic approaches for studying the roles of intermolecular H-bonds and supramolecular structures in the mechanisms of homogeneous catalysis and enzymatic catalysis on model systems.

## Data Availability

All of the experimental data presented belong to the authors of this manuscript and are available.

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
