# Peer review of "The Role H-Bonding and Supramolecular Structures in Homogeneous and Enzymatic Catalysis"

_ijms, 2023, doi:10.3390/ijms242316874_

Round 1
Reviewer 1 Report
Comments and Suggestions for Authors
The manuscript “The Role H-bonding and Supramolecular Structures in Mechanisms Homogeneous and Enzymatic Catalysis” by L. I. Matienko, E. M. Mil, A. A. Albantova and A. N. Goloshchapov is an interesting review on the non-covalent interactions as key factors in the catalysis. This topic is not yet fully explored and the review is indeed timely, especially as the authors have in the past proposed an AFM-based protocol for studying self-aggregation and formation of nanoparticles. I recommend the publication of this manuscript in the Special Issue “Recent Advances in Hydrogen Bonding” of the International Journal of Molecular Sciences; however, there is one large issue which should be resolved, described immediately below.
I am not a native English speaker, so I do not consider myself as competent to assess the quality of the writing, however I recognize that the manuscript requires linguistic attention. There are numerous grammar omission, especially lack of the preposition “of” – even in the title, which contains the phrase “in Mechanisms Homogeneous and Enzymatic Catalysis”; I would rather read “ in Mechanisms of Homogeneous and Enzymatic Catalysis”. Also for example, the line 10 (abstract) ends with “analyzed”, while I think “is analyzed” would be more appropriate. Such omissions are frequent, and indeed these linguistic issues are the only factor restraining me from recommending immediate publication.
The authors say that „this review” is prepared „with an emphasis on the authors' works”, however self-citations do not exceed 20% in my estimation, which is good indication that the review is indeed thorough and covers relevant studies of diverse groups. The choice of the analyzed systems is significant: the authors describe the “second coordination sphere” notion and analyze the role of H-bonds using diverse examples of nickel and iron catalysts. The described catalysts are not only forming supramolecular structures; some of them are bio-mimetic, emulating the active centers of the P450 enzymes. This is another reason for my high evaluation of the overall merit of the manuscript.
Smaller issues:
The list of abbreviations is only partially ordered – it seems that the intent was to make it alphabetical, but MSt and His are not in place.
There are some errors in references, e.g. Ref. 12: the names “Grabovski, Krygovski” should be “Grabowski, Krygowski”; Ref. 29 and 35: some spaces are misplaced.
End of reviewer report
Comments on the Quality of English Language
I am not a native English speaker, so I do not consider myself as competent to assess the quality of the writing, however I recognize that the manuscript requires linguistic attention. There are numerous grammar omission, especially lack of the preposition “of” – even in the title, which contains the phrase “in Mechanisms Homogeneous and Enzymatic Catalysis”; I would rather read “ in Mechanisms of Homogeneous and Enzymatic Catalysis”. Also for example, the line 10 (abstract) ends with “analyzed”, while I think “is analyzed” would be more appropriate. Such omissions are frequent, and indeed these linguistic issues are the only factor restraining me from recommending immediate publication.
Author Response
please look at the attachments

Reviewer 2 Report
Comments and Suggestions for Authors
Manuscript ID: ijms-2721302
Title: The Role H-bonding and Supramolecular Structures in Mechanisms Homogeneous and Enzymatic Catalysis.
Authors: Ludmila I. Matienko *, Elena M. Mil, Anastasia A. Albantova, Alexander N. Goloshchapov
Comments:
1. What is the main question addressed by the research?
This minireview addresses the importance of the second coordination sphere for the catalytic activity of the active site of organometallic complexes. In other words, the importance of moving from the simplified metal-ligand model to considering a more accurate model based on the formation of supramolecular structures is emphasized. The use of three-dimensional atomic force microscopy as one of the promising experimental methods for studying such supramolecular structures is highlighted and discussed.
2. Do you consider the topic original or relevant in the field? Does it address a specific gap in the field?
I fully support the authors in emphasizing the importance of considering the influence of the environment on the reactivity of catalysts. This aspect is often not discussed due to the complexity of its research and accounting. I would like to recommend that the authors mention a similar problem of the influence of multiple weak interactions on the structure of complexes with a single strong hydrogen bond. This topic was studied in more detail, which made it obvious that the total energy of these weak interactions can exceed the energy of the main interaction (see e.g. DOI: /10.1524/zpch.2013.0400 and other publications on this topic).
3. What does it add to the subject area compared with other published material?
The mini-review gives an idea of the actual capabilities of atomic force microscopy in the study of self-organizing supramolecular structures involving organometallic complexes.
4. What specific improvements should the authors consider regarding the methodology? What further controls should be considered?
Lines 100-102. The partially covalent nature of strong hydrogen bonds is experimentally demonstrated by NMR spectroscopy. Electron-mediated couplings between nuclei on both sides of the hydrogen bridge discovered in biological macromolecules (see e.g. DOI: 10.1007/0-306-47936-2_9) can be quite large in model molecular systems (see e.g. DOI: 10.1002/(SICI)1521-3765(19990201)5:2<492::AID-CHEM492>3.0.CO;2-I).
Lines 377-382: Correct the flow and logic of the text.
In general, use J/mol instead of kcal/mol.
5. Are the conclusions consistent with the evidence and arguments presented and do they address the main question posed?
Yes.
6. Are the references appropriate?
Minor improvements are suggested above.
7. Please include any additional comments on the tables and figures.
I have no comments on the figures and tables that could improve the article.
Author Response
please look at the attachments
